# Evolving Role of [^18^F]Flurocholine PET/CT in Assessing Primary Hyperparathyroidism: Can It Be Considered the First-Line Functional Imaging Approach?

**DOI:** 10.3390/jcm12030812

**Published:** 2023-01-19

**Authors:** Seyedeh Sara Seyedinia, Seyed Ali Mirshahvalad, Gregor Schweighofer-Zwink, Lukas Hehenwarter, Gundula Rendl, Christian Pirich, Mohsen Beheshti

**Affiliations:** 1Division of Molecular Imaging & Theranostics, Department of Nuclear Medicine, University Hospital Salzburg, Paracelsus Medical University, 5020 Salzburg, Austria; 2Joint Department of Medical Imaging, University Health Network, University of Toronto, Toronto, ON M5G 2M2, Canada

**Keywords:** parathyroid, PET, CT, [^18^F]fluorocholine, PTH, calcium

## Abstract

The accurate detection of hyperfunctioning parathyroid tissue (HFPT) is pivotal in the preoperative assessment of primary hyperparathyroidism (PHPT). PET/CT using [^18^F]fluorocholine ([^18^F]FCH) showed superior diagnostic performance compared to conventional functional imaging modalities. We aimed to evaluate the diagnostic performance of [^18^F]FCH PET/CT as a first-line functional imaging approach in patients with clinically diagnosed PHPT. The imaging and clinical data of 321 PHPT patients, including 271 overt PHPT and 50 mild PHPT, who underwent [^18^F]FCH PET/CT as first-line imaging were analysed in this retrospective study. Histopathology was the reference standard. In case of no available histopathology evaluation (conservative management), imaging and clinical follow-ups were considered reference standards. In the overt group (*n* = 271), [^18^F]FCH PET/CT showed sensitivity, specificity, positive predictive value (PPV), negative predictive value (NPV), and accuracy of 0.99, 0.91, 1.00, 0.80, and 0.99, respectively. Regarding the correlation of the index lesions and initial laboratory data, all [^18^F]FCH PET/CT parameters (SUVs, SULs, and mSAD) were significantly correlated with the serum iPTH level. Additionally, SUVmax, SULpeak, and mSAD were significantly associated with the serum calcium level. In the mild group (*n* = 50), [^18^F]FCH PET/CT showed a sensitivity, specificity, PPV, NPV, and accuracy of 0.93, 0.75, 0.95, 0.67, and 0.90. In conclusion, [^18^F]FCH PET/CT revealed high diagnostic performance in the detection of HFPTs and the potential to be considered as a first-line imaging modality in the assessment of PHPT, including both overt and mild types. However, its cost–benefit concerning the clinical impact of early PHPT detection should be investigated in future studies.

## 1. Introduction

Primary hyperparathyroidism (PHPT) is the third most common endocrine disorder. It is characterized by elevated or inappropriately normal parathyroid hormone serum (PTH) levels due to single or multiple hyperactive parathyroid glands [1]. Parathyroid adenoma is the most common type of PHPT, followed by multi-glandular disease (MGD), parathyroid hyperplasia, and rarely parathyroid carcinoma [1].

The *overt PHTP* (oPHPT) is defined as an elevated serum level of calcium following an increase in PTH, leading to further clinical symptoms [2,3,4]. This is the classic definition of PHPT, which has been considered clinically significant for decades. However, routine determination of the serum calcium as a biochemical-screening tool resulted in the identification of asymptomatic patients with borderline biochemical results [5]. Thus, the term *mild PHPT* (mPHPT) was introduced. mPHPT is considered a different entity regarding histopathology and management, and has been categorized into two different groups, *Normohormonal mPHPT* (NHmPHPT) and *Normocalcemic mPHPT* (NCmPHPT) [6,7]. Approximately half of mPHPT patients and two-thirds of NHmPHPT cases experience at least one non-specific symptom, only to be recognized upon questioning [5,8].

Surgical removal of parathyroid adenoma(s) is the curative approach for PHPT [9]. Minimally invasive parathyroidectomy (MIP) is currently the procedure of choice with fewer morbid events compared to the traditional bilateral neck exploration (BNE) [10,11]. Precise preoperative localization of the hyperfunctioning parathyroid tissue(s) (HFPT) is necessary to perform MIP and may prevent persistent/recurrent PHPT and re-operation. Noteworthy, considering only mPHPT patients, decision-making is more challenging since surgery is still controversial, especially in asymptomatic patients [5]. Additionally, hesitation still exists among surgeons because of the higher surgical failure compared to oPHPT (5% vs. 1%), due to the higher incidence of multiglandular disease (as high as 35%) and smaller size of the adenoma(s) [5,7,12,13,14,15].

Cervical ultrasonography (cUS) and parathyroid scintigraphy with [^99m^Tc]Tc-Methoxy-IsoButyl-Isonitrol ([^99m^Tc]Tc-MIBI) is the first-line imaging modalities for preoperative detection and localization of HFPT [16]. However, cUS is an operator-dependent modality with limited diagnostic performance in the presence of small adenomas and ectopic glands [17]. Considering [^99m^Tc]Tc-MIBI, in a meta-analysis by Wei et al., single photon emission computed tomography/computed tomography (SPECT/CT) and SPECT had sensitivities of 0.84 and 0.66, respectively [18]. [^99m^Tc]Tc-MIBI has also revealed limited performance, particularly in the presence of small parathyroid adenoma or MGD [19]. Furthermore, anatomical imaging, including CT and magnetic resonance imaging (MRI), revealed limited impact in assessing PHPT. The reported sensitivities of CT for the detection of HFPTs range 0.46–0.87. Additionally, the diagnostic performance of MRI for accurate identification of HFPTs is highly dependent on their typical morphology and location [20].

^18^F-flurocholine ([^18^F]FCH) is an analogue of phospholipid that binds to a new synthetised membrane. HFPT functioning parathyroid tissues upregulate the expression of ^2+^Calcium-dependent choline kinase receptors, resulting in increased [^18^F]FCH uptake [21]. Furthermore, [^18^F]FCH PET/CT provides higher spatial resolution enabling the detection of smaller lesions [11,22]. In our previous prospective dual-centre study, we found the superiority of [^18^F]FCH PET/CT in the detection and localization of parathyroid adenomas compared to [^99m^Tc]Tc-MIBI SPECT/CT as a standard of care imaging [11]. Meanwhile, several other studies have demonstrated similar results. Thanks to the available PET/CT scanners and radiopharmaceutical facilities nowadays, the question is raised if [^18^F]FCH PET/CT can be performed as the first-line preoperative imaging modality. Thus, in the current study, we aimed to evaluate the diagnostic performance of [^18^F]FCH PET/CT as the first-line preoperative imaging in a large patient population with clinically diagnosed PHPT.

## 2. Materials and Methods

### 2.1. Study Population

This IRB-approved study was performed in accordance with the ethical standards of the institutional ethics committee and with the 1964 Helsinki declaration and its later amendments or comparable ethical standards.

Overall, 360 consecutive patients with the initial diagnosis of PHPT who underwent [^18^F]FCH PET/CT from January 2015 to December 2021 were included in this study. Exclusion criteria were the initial diagnosis of secondary/tertiary hyperparathyroidism (*n* = 11), as well as vitamin D deficiency (*n* = 4), and familial hypocalciuric hypercalcemia (*n* = 4) that were diagnosed after further investigation. Moreover, patients without reliable follow-up clinical or imaging data (*n* = 19) and poor quality of the [^18^F]FCH PET/CT imaging (*n* = 1) were also excluded. To precisely evaluate our cohort and prevent possible biases, we divided the patients into two groups: oPHPT (*n* = 271) and mPHPT (*n* = 50). The definitions and overall characteristics are provided in the supplemental materials. oPHPT was defined as elevated serum levels of both iPTH and calcium. Patients with the diagnosis of mPHPT were: (a) *NCmPHPT*: patients with normal serum calcium levels and elevated serum intact iPTH levels; and (b) *NHmPHPT*: patients with normal serum iPTH levels and elevated serum calcium levels.

### 2.2. [^18^F]FCH PET/CT Acquisition and Interpretation

Detailed acquisition and image interpretation protocols are provided in the supplemental materials. The number and location of the abnormal lesions were noted. The location of parathyroid lesions was categorized into two groups, upper parathyroid glands (P4) and lower parathyroid glands (P3) [23]. These regions were correlated with the surgical resection reports to find the corresponding gland. The semi-quantitative parameters, including maximum standard uptake value (SUVmax), average SUV (SUVmean), peak and average SUV normalized by lean body mass (SULpeak and SULmean, respectively), were measured within a selected iso-contour sphere as the volume of interest (VOI), with a predefined threshold of 41%. The maximum metabolic short-axis diameters (mSAD) of the abnormal parathyroid lesion were determined on the PET images.

### 2.3. Reference Standard

The final diagnosis was based on histopathological results in patients who underwent surgery (all underwent the MIP procedure, except for three). The interval between PET/CT imaging and surgery was no longer than one month (mean days interval = 21 ± 8). A successful resection was considered as >50% intraoperative iPTH decline compared to the baseline iPTH and decreasing pattern of iPTH and calcium in the laboratory follow-ups within six months after surgery. Pathologic parathyroid specimens were reported as adenoma, hyperplasia, or carcinoma. In patients with negative parathyroid pathology, the presence of thymus tissue, thyroid nodules, or lymph nodes was extracted. Histopathological results were considered the reference standard for the resected glands. Additional information was gathered from the pathology reports, including the size and weight of the parathyroid lesions.

In the group of oPHPT patients who underwent the conservative approach, imaging (e.g., follow-up US or [^99m^Tc]Tc-MIBI imaging) and/or clinical (laboratory evaluation) follow-ups were considered the reference standard. A 6-month follow-up included laboratory data (serum level of iPTH and calcium) and cUS. In patients with MGD, the lesion with the highest [^18^F]FCH uptake was defined as index-lesion for the analyses.

Clinical and imaging follow-ups were considered the reference standard in the group of mPHPT patients who underwent the conservative approach. In the NCmPHPT and NHmPHPT patients, normalized iPTH and calcium or a more than 50% decrease in iPTH from the baseline value was considered significant.

The detailed definitions of true-positive (TP), true-negative (TN), false-positive (FP), and false-negative (FN) results can be found in the Appendix A.

### 2.4. Statistical Analysis

Continuous variables were expressed as mean values with standard deviations (SD). Non-parametric *t*-test was applied to determine the significance of differences for continuous variables. Categorical variables were expressed as number and frequency. Diagnostic performances were calculated, including sensitivity, specificity, positive predictive value (PPV), negative predictive value (NPV), and accuracy. According to the reference standard, the receiver operating characteristic (ROC) curves fitted for sensitivity and specificity were drawn. Additionally, the area under the curve (AUC) was calculated. All statistical analyses were performed using SPSS (ver.26, IBM) software. A two-tailed *p*-value of <0.05 was considered statistically significant.

## 3. Results

### 3.1. oPHPT Population

#### Patient-Based Analysis

In 271 patients, [^18^F]FCH PET/CT showed TP, TN, FP, and FN results in 257, 10, 1, and 3 patients, respectively. Table 1 provides details about the included patients. Table 2 shows detailed initial and follow-up laboratory results of the included patients.

In the patient-based analyses, [^18^F]FCH PET/CT showed sensitivity, specificity, positive predictive value (PPV), negative predictive value (NPV), and accuracy of 0.99, 0.91, 1.00, 0.80, and 0.99, respectively.

Overall, 24 (24/271, 9%) patients revealed abnormal [^18^F]FCH uptake in more than one location (13 patients with two, 9 patients with three, and 2 patients with four lesions) suggestive of MGD (Figure 1). In this group, 14 patients underwent the surgical approach, and 10 followed up with conservative management. No significant differences were found between laboratory parameters of patients with MGD compared to single glandular disease (*p*-values > 0.05).

There were three patients with evidence of persistent iPTH or Calcium on follow-up. All these patients had positive [^18^F]FCH PET/CT and pathology results. After a retrospective review of the [^18^F]FCH PET/CT images in one of them, we noticed evidence of additional mild small radiotracer uptake suggestive of parathyroid adenoma that could be the reason for persistent laboratory results. Two other patients had positive [^18^F]FCH PET/CT results, but the available pathology reports were negative for HFPT. These patients showed a more than 50% decrease in iPTH during the surgery, and iPTH remained in the normal range years after the surgery.

Three patients in the FN group had negative [^18^F]FCH PET/CT results and increased levels of iPTH (*n* = 2) or Calcium (*n* = 1) on the follow-up. Only one patient was considered as FP. In this case, [^18^F]FCH PET/CT localized one lesion suggesting HFPT, and subsequent surgery was performed. However, the pathology report was negative for HFPT. The iPTH serum levels showed initially <50% decline after surgery but an increasing pattern in the clinical follow-ups. No information was available about intraoperative iPTH assessment in this case, as the surgery was performed in another hospital.

### 3.2. Lesion-Based Analysis

Metabolic characteristics (mean ± SD) SUVmax, SUVmean, SULpeak, SULmean, and mSAD of index lesions (*n* = 257) were 9.1 ± 3.6, 5.3 ± 1.8, 6.3 ± 2.5 and 3.7 ± 1.3, and 8 ± 5 mm, respectively. Moreover, histology characteristics, including size and weight of the positive lesions, were 17 ± 7 mm and 1.2 ± 1.7 g, respectively. From a total of 309 pathologic lesions, 231 were considered P4, and 78 were P3. Eighteen patients (7%) had ectopic glands, including fourteen P4 lesions and four P3 lesions. Thirteen ectopic P4 lesions were identified in the mediastinum and one in the paravertebral region. Three P3 ectopic glands were ectopic cervical adenomas, and one was an intrathyroidal adenoma. Figure 2 shows a patient with pathology-proven ectopic parathyroid adenoma.

Regarding the correlation of the index lesions and initial laboratory data, all [^18^F]FCH PET/CT parameters (SUVs, SULs, and mSAD) were significantly correlated with the serum iPTH level. Additionally, SUVmax, SULpeak, and mSAD were significantly associated with the serum calcium level. Details are provided in Table 3. Other laboratory parameters, including phosphorus, alkaline phosphatase, and creatinine, showed no significant correlation with the semi-quantitative findings on [^18^F]FCH PET/CT (*p*-values > 0.05). Furthermore, there was a significant correlation between SUV values on [^18^F]FCH PET and histopathology reports, including size and weight (*p*-values < 0.05). SUVmean was the only semi-quantitative parameter of [^18^F]FCH PET/CT, which was significantly different between parathyroid adenomas and hyperplasia (*p*-value < 0.05, Figure 3). However, based on clinical and imaging findings, accurate differentiation between various HFPTs was not possible.

## 4. mPHPT Population

A total number of 50 patients (mean age= 67 ± 12 years) was studied, including 39 (78%) patients with NCmPHPT and 11 (22%) patients with NHmPHPT disease. Detailed characteristics are provided in Table 1. In the NCmPHPT group, 32/39 (82%) patients showed positive results on [^18^F]FCH PET/CT images. In the NHmPHPT group, 9/11 (82%) patients were positive. There were no significant differences in semi-quantitative parameters between NCmPHPT and NHmPHPT patients (*p*-values > 0.05). Out of 39 NCmPHPT patients, 19 (49%) underwent surgical resection of the positive [^18^F]FCH PET/CT gland. Detailed histopathology findings can be found in Table 4. In the NHmPHPT group, 4/11 (36%) patients were planned for surgery. MIP was performed for all patients except three who underwent BNE; two were NCmPHPT, and one was in the NHmPHPT group. Pathology found at least one parathyroid adenoma in all patients with positive scans, except for an NCmPHPT patient with only one hyperplastic gland on pathology. In this patient, iPTH decreased more than 50% at the 6-month follow-up. In the remaining patients, including 20 NCmPHPT and 7 NHmPHPT patients, follow-up laboratory results were considered the reference standard. In the NCmPHPT patients, 11, 4, 2, and 3 were categorized in TP, TN, FP, and FN results, respectively. In the NHmPHPT patients, follow-up revealed 5 TP and 2 TN cases. Overall, [^18^F]FCH PET/CT showed a sensitivity, specificity, PPV, NPV, and accuracy of 0.93, 0.75, 0.95, 0.67, and 0.90.

Since this group of patients was more controversial in terms of management, except for one NCmPHPT case, all patients underwent cUS examination. A comparison of findings between [^18^F]FCH PET/CT, cUS, and the final diagnosis is provided in Table 5. We analysed the [^18^F]FCH PET/CT versus cUS performances in these 49 patients. In NCmPHPT patients, sensitivity, specificity, and accuracy of [^18^F]FCH PET/CT vs. cUS were 0.91 vs. 0.38, 0.67 vs. 0.83, and 0.84 vs. 0.45, respectively. In NHmPHP group, sensitivity, specificity, and accuracy of [^18^F]FCH PET/CT vs. cUS were 1.00 vs. 0.44, 1.00 vs. 0.50, and 1.00 vs. 0.45, respectively. ROC curves were drawn for [^18^F] FCH PET/CT vs. cUS, showing AUCs of 0.84 vs. 0.57 (Figure 4).

## 5. Discussion

[^18^F]FCH PET/CT holds promising diagnostic performance for becoming the first-line imaging modality for accurate detection and localization of HFPT in PHPT patients. To our knowledge, this study assessed the impact of [^18^F]FCH PET/CT as first-line imaging in the largest patient population to date, including both oPHPT and mPHPT. The results of the current study can support the high diagnostic performance of [^18^F]FCH PET/CT in oPHPT with a sensitivity, specificity, PPV, NPV, and accuracy of 0.99, 0.91, 1.00, 0.80, and 0.99, respectively. Additionally, it showed high accuracy in mPHPT patients. Interestingly, [^18^F]FCH PET/CT could detect and localize HFPTs in all NHmPHPT patients accurately.

Localization of HFPT is mandatory before resection to reduce the risk of further surgical failure [24]. Based on the findings of our initial prospective research in PHPT patients comparing [^18^F]FCH PET/CT with [^99m^Tc]Tc-MIBI SPECT/CT, [^18^F]FCH PET/CT showed superiority to be the first-line functional imaging modality in our centre [11]. We reported detection rates of 0.93 and 0.61 for [^18^F]FCH PET/CT and [^99m^Tc]Tc-MIBI SPECT/CT, respectively. However, in most centres worldwide, [^99m^Tc]Tc-MIBI SPECT/CT is still considered the modality of choice following current clinical guidelines for the localization of HFPT before the surgery [11,25,26,27,28,29,30]. Regarding its limitations, [^99m^Tc]Tc-MIBI scintigraphy provided low sensitivity and high numbers of FN results [11,24,31]. Moreover, its sensitivity was even lower in specific conditions, including MGD and ectopic adenoma [11,16,24,31]. Danovan et al. reported an overall sensitivity of 0.58 [32]. While in the presence of solitary adenoma, the sensitivity was 0.78, it dropped to 0.31 in MGD patients [32]. Another important shortcoming of [^99m^Tc]Tc-MIBI was the detection of small adenomas, showing a limited accuracy in small HFPTs with a median weight of 0.5 g [31]. In addition, the diagnostic performance of [^99m^Tc]Tc-MIBI was very limited in NCmPHPT patients in the study by Gomez- Ramirez et al. [33]. They proposed the hypothesis that the lower level of iPTH decreases the [^99m^Tc]Tc-MIBI diagnostic capability. Thus, considering mPHPT as a similar entity to oPHPT, may not be a ground truth from the imaging perspective. Furthermore, although the combination of planar and SPECT/CT parathyroid scintigraphy increased the sensitivity up to 0.86 in PHPT patients, again, [^99m^Tc]Tc-MIBI suffered from acceptable diagnostic performance in mPHPT patients [13,34]. Musumeci et al. reported a sensitivity of 0.59 for [^99m^Tc]Tc-MIBI SPECT/CT in mPHPT compared to 0.86 in oPHPT, which was similar to Gomez-Ramirez et al. [10,33]. Additionally, [^18^F]FCH PET/CT provides higher spatial resolution and lesion to background ratio [9], semi-quantitative assessment, shorter imaging time [11,31], and lower radiation dose (2.8 mSv vs. 6.8 mSv) compared to [^99m^Tc]Tc-MIBI SPECT/CT [22,35,36]. Moreover, there is no need to discontinue calcimimetic drugs before [^18^F]FCH PET/CT [22].

Our previous study revealed that [^99m^Tc]Tc-MIBI SPECT/CT imaging was falsely negative in adenomas <13 mm [11]. Similar findings were reported in other studies, showing that [^99m^Tc]Tc-MIBI SPECT/CT were mostly negative in HFPT ≤12mm [37,38]. In the current study, 61% (130/241) of surgically removed HFPT were reported to be ≤15 mm on the histopathology. [^18^F]FCH PET/CT correctly detected all of these small lesions, which can support the potential of this modality in detecting small HFPTs.

Several previously published studies have evaluated [^18^F]FCH PET/CT as the first-line imaging modality in PHPT. Cuderman et al. showed that [^18^F]FCH PET/CT is a diagnostic modality superior to conventional imaging [31]. Wouter et al. assessed [^18^F]FCH PET/CT as an exclusive imaging modality and reported a detection rate of 96% on per-patient and 90% on per-lesion analysis in 139 PHPT patients who underwent surgery [22]. Although the initial population in this study was larger, the final analysis was performed only in a subgroup that underwent surgery. Bossert et al. recently reported [^18^F]FCH PET/CT as an effective imaging modality in NCmPHPT [39]. Notably, they also indicated [^18^F]FCH PET/CT as a cost-effective modality due to its high diagnostic performance and accurate pre-surgical localization of HFPTs. However, they compared oPHPT (*n* = 27) with limited NCmPHPT patients (*n* = 7) and did not evaluate NHmPHPT. In contrast, in the current study, we tried to assess the diagnostic performance of [^18^F]FCH PET/CT in a larger population and evaluated NHmPHPT patients as well.

The impact of [^18^F]FCH PET/CT in detecting MGD and ectopic parathyroid adenomas has been discussed in the literature. Most studies reported promising results emphasizing the potential of [^18^F]FCH PET/CT to overcome the limitations of other modalities, as well as the determination of the lipomatous parathyroid adenomas [16,26,40,41]. In contrast, some limited studies revealed a probability of FP findings on [^18^F]FCH PET/CT imaging [31,42]. Our result may support that [^18^F]FCH PET/CT is an accurate modality in MGD patients. This was of great clinical importance considering the limited sensitivity of the intraoperative iPTH approach in MGD, which may lead to premature surgical termination, prolongation of the surgery time, and an increase in the total costs [31,43]. Of note, the retrospective nature of our study might influence our results.

Few previous studies assessed the correlation between metabolic parameters derived by [^18^F]FCH PET/CT and serum levels of iPTH and calcium. Alharbi et al. found a strong correlation between SUVmax of the positive lesion and serum level of iPTH [44]. Other studies reported a strong correlation between serum levels of calcium and SUVmax [30,45]. We also found strong correlations between all semi-quantitative metabolic parameters and serum iPTH levels. In terms of the serum calcium level, SUVmax and SULpeak showed a significant correlation in the current study.

One of the limitations of our study was the lack of histopathologic confirmation in patients who underwent the conservative approach. Although the laboratory results could characterize patients to some extent, histopathology is the gold standard for a definite diagnosis. Additionally, the retrospective design of the study could affect the calculated performances. Moreover, only a limited number of genetic test results were available in medical records, and urinary calcium measurement was performed only in a few patients with clinically suspicious familial hypocalciuric hypercalcemia. Therefore, providing a reliable analysis was not possible. Lastly, the number of mPHPT patients, especially in the NHmPHPT group, was limited.

## 6. Conclusions

In conclusion, this study revealed a high diagnostic performance of [^18^F]FCH PET/CT in detecting HFPTs in both oPHPT and mPHPT patients. Thus, adding our results to the current literature, [^18^F]FCH PET/CT can be considered as first-line imaging for assessing PHPT in terms of diagnostic accuracy. Additionally, preliminary studies showed [^18^F]FCH PET/CT to be a potentially cost-effective modality in this regard [39,46], though further cost–utility investigations are mandatory to recommend [^18^F]FCH PET/CT as a routine modality in the clinic.

## Figures and Tables

**Figure 1 jcm-12-00812-f001:**
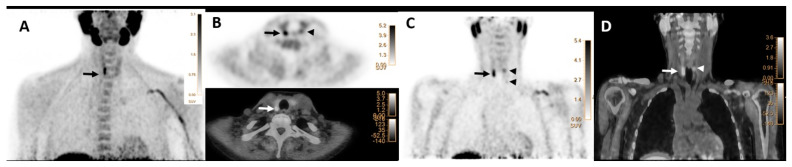
[^18^F]FCH PET/CT images (**A**–**D**) in a 67-year-old woman with elevated serum PTH of 102 pg/mL (normal value = 16–65) and calcium level of 2.67 mmol/l (normal value = 2.13–2.63), respectively. (**A**) Maximum intensity projection (MIP) shows an elongated focus of tracer uptake cranial to the right thyroid lobe (arrow). (**B**) transaxial PET (upper row) and hybrid PET/CT (lower row) images show focal tracer uptake posterior to the lower pole of the right thyroid lobe with SUV max of 8.3, mSAD = 5.5mm (arrows). Another mild focal uptake, posterior to the left thyroid upper-pole with SUVmax of 3.5 and mSAD = 3.5mm (arrowheads). (**C**) Coronal PET image shows two foci of small abnormal tracer uptake posterior to the upper poles of the thyroid (arrow and superior arrowhead) and one additional small faint uptake (inferior arrowhead) posterior to the lower pole of the left thyroid (SUVmax = 1.85 and mSAD = 2.68mm). (**D**) The corresponding coronal hybrid PET/CT image shows the exact localization of the detected lesions (arrows). The findings are suggestive of multiglandular parathyroid disease verified as three parathyroid adenomas on histopathology.

**Figure 2 jcm-12-00812-f002:**
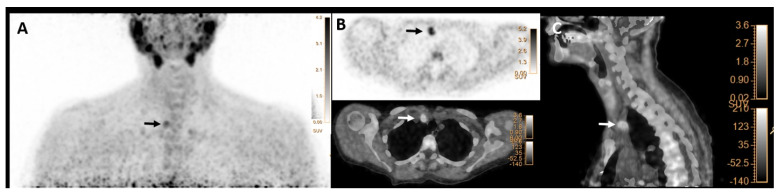
[^18^F]FCH PET/CT images (**A**–**C**) in a 62-year-old woman with elevated serum PTH of 190 pg/mL (normal value = 16–65) and calcium level of 2.89 mmol/l (normal value = 2.13–2.63), respectively. (**A**) Maximum intensity projection (MIP) shows focal tracer uptake in the right upper mediastinum (arrow). (**B**) transaxial PET (upper row) and hybrid PET/CT (lower row) as well as (**C**) sagittal hybrid PET/CT images show focal tracer uptake in the right upper mediastinal region, lateral to the truncus brachiocephalicus with SUVmax of 5.2 and mSAD = 11mm, compatible with an ectopic parathyroid adenoma (arrows), that was found to be 20mm ectopic parathyroid adenoma on histopathology result.

**Figure 3 jcm-12-00812-f003:**
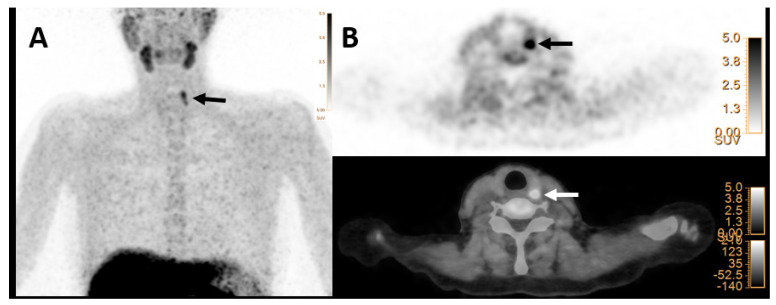
[^18^F]FCH PET/CT images (**A**,**B**) in a 53-year-old woman with elevated serum PTH of 73 pg/mL (normal value = 16–65) and calcium level of 2.70 mmol/L (normal value = 2.13–2.63), respectively. (**A**) Maximum intensity projection (MIP) shows focal tracer uptake in the cranial part of the left thyroid lobe (arrow). (**B**) Transaxial PET image (upper row) and hybrid PET/CT (lower row) images show focal tracer uptake posterior to the upper pole of the left thyroid lobe with SUVmax of 8.2 and mSAD = 8.2mm, which was found to be hyperplasia on histopathology result.

**Figure 4 jcm-12-00812-f004:**
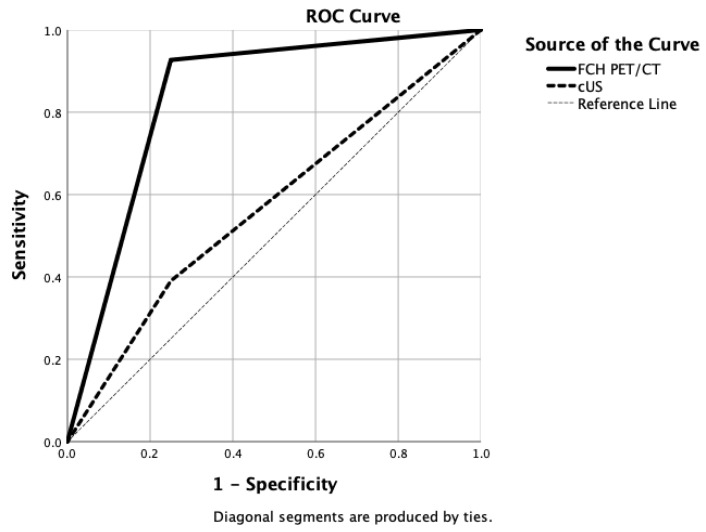
Drawn ROC curves for [^18^F]FCH PET/CT (AUC = 0.84) versus clinical ultrasonography (AUC = 0.57) diagnostic accuracies in detecting mild primary hyperparathyroidism.

**Table 1 jcm-12-00812-t001:** Detailed characteristics of the patient’s population (*n* = 271).

	Mean ± SD|Number (%)		
Patient Diagnosis	oPHPT	NCmPHPT	NHmPHPT
Age	66 ± 13	68 ± 13	62 ± 11
BMI	27 ± 6	27 ± 5	26 ± 4
iPTH (pg/dL)	159 ± 160	135 ± 96	58 ± 14
Calcium (mmol/L)	2.8 ± 0.2	2.38 ± 0.27	2.74 ± 0.05
Phosphorus (mmol/dL)	0.8 ± 0.2	0.83 ± 0.18	0.85 ± 0.23
Creatinine (mg/dL)	1.2 ± 1.6	0.9 ± 0.3	1 ± 0.6
Alkaline phosphatase (U/I)	88 ± 37	76 ± 27	74 ± 16
Female/Male	225/46 (83%/17%)	32/7 (82%/18%)	7/4 (64%/36%)
History of thyroid disease	172 (64%)	26 (67%)	7 (64%)
History of renal stone	28 (10%)	3 (8%)	1 (9%)
History of cholelithiasis	18 (7%)	2 (5%)	1 (9%)
Surgery/Conservative	200/71 (74%/26%)	19/20 (49%/51%)	4/7 (36%/64%)

**Table 2 jcm-12-00812-t002:** Detailed laboratory results of the included patients.

		Initial PTH (pg/dL)	Follow-Up PTH (pg/dL)	Initial Calcium (mmol/L)	Follow-up Calcium (mmol/L)
True-Positive	Overt	164 ± 172	62 ± 58	2.78 ± 0.27	2.43 ± 0.33
	Normocalcemic	142 ± 105	76 ± 57	2.36 ± 0.31	2.42 ± 0.15
	Normohormonal	62 ± 8	58 ± 6	3.11 ± 0.18	2.40 ± 0.22
True-Negative	Overt	180 ± 228	78 ± 47	2.79 ± 0.21	2.62 ± 0.17
	Normocalcemic	77 ± 6	42.75 ± 12.40	2.33 ± 0.06	2.42 ± 0.09
	Normohormonal	41 ± 28	48 ± 24	2.91 ± 0.01	2.86 ± 0.02
False-Positive	Overt	91	103	2.71	2.56
	Normocalcemic	118 ± 21	107 ± 18	2.50 ± 0.04	2.52 ± 0.30
	Normohormonal	-	-	-	-
False-Negative	Overt	103 ± 1	117 ± 0.7	2.72 ± 1	2.77 ± 1
	Normocalcemic	160 ± 81	168 ± 109	2.47 ± 0.03	2.51 ± 0.25
	Normohormonal	-	-	-	-

Data represented as mean values ± SD.

**Table 3 jcm-12-00812-t003:** Correlation between the semi-quantitative parameters of the index lesions on [^18^F]FCH PET/CT and initial laboratory data.

[^18^F]FCH PET/CT Parameters	Overt PHPT		Normocalcemic mPHPT		Normohormonal mPHPT	
	iPTH (r_p_ *, *p*-Value)	Calcium (r_p_, *p*-Value)	iPTH (r_p_, *p*-Value)	Calcium (r_p_, *p*-Value)	iPTH (r_p_, *p*-Value)	Calcium (r_p_, *p*-Value)
SUVmax	0.199, <0.01	0.163, <0.01	0.234, >0.1	−0.353, <0.05	−0.055, >0.1	−0.699, <0.05
SUVmean	0.214, <0.01	0.092, >0.1	0.364, <0.05	−0.171, >0.1	−0.090, >0.1	−0.586, >0.05
SULpeak	0.237, <0.001	0.187, <0.01	0.299, >0.1	−0.234, >0.1	−0.017, >0.1	−0.706, <0.05
SULmean	0.190, <0.01	0.076, >0.1	0.305, >0.05	0.064, >0.1	−0.105, >0.1	−0.686, <0.05
mSAD (mm)	0.218, <0.001	0.134, <0.05	0.364, <0.05	−0.201, >0.1	−0.845, <0.01	−0.360, >0.1

* Pearson’s r.

**Table 4 jcm-12-00812-t004:** Detailed histopathology findings of the removed glands.

	Pathology				
	Adenoma	Hyperplasia	Normal	Size	Weight
Overt PHPT	201	5	8	17 ± 7 (mm)	1.2 ± 1.7 (g)
Mild PHPT	22	1	0	14.4 ± 6.2 (mm)	0.75 ± 0.8 (g)

**Table 5 jcm-12-00812-t005:** Detailed patient-based findings of mPHP-cohort in [18F] FCH PET/CT and clinical ultrasonography (cUS) versus the reference standard.

Reference Standard	NCmPHP (*n* = 38)				NHmPHP (*n* = 11)			
	FCH PET/CT		cUS		FCH PET/CT		cUS	
	+	−	+	−	+	−	+	−
Positive (+)	29	3	12	20	9	0	4	5
Negative (−)	2	4	1	5	0	2	1	1

## Data Availability

The data presented in this study are available on request from the corresponding author.

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
