# Peer review of "Evolving Role of [18F]Flurocholine PET/CT in Assessing Primary Hyperparathyroidism: Can It Be Considered the First-Line Functional Imaging Approach?"

_jcm, 2023, doi:10.3390/jcm12030812_

Round 1

Reviewer 1 Report

General comments

The authors retrospectively review a large patient cohort of patients with primary hyperparathyroidism (pHPT) that underwent [18F]fluorocholine ([18F]FCH) PET/CT as a first-line imaging for localization of hyperfunctioning parathyroid tissue. The patients were subdivided into the “overt” and “mild” pHPT categories according to symptoms and laboratory results; the “mild” pHPT group was further subdivided into the “normocalcemic” and “normohormonal” categories. Diagnostic peroformance of [18F]FCH PET/CT was evaluated against post-op histology as a standard of truth; interestingly, a number of patients was managed conservatively without referral to surgery with F/U w/imaging and lab data as a standard of truth. Imaging-derived metabolic parameters were correlated to laboratory results. [18F]FCH PET/CT was shown to have an impressive diagnostic performance in both “overt” and “mild” pHPT groups with the derived metabolic parameters significantly correlated to some biochemistry data.

Overall, the study confirms the already known excellent diagnostic performance of [18F]FCH PET/CT in pHPT; the relation between biochemistry values and (most of) the metabolic metrics is in keeping with several previous works (and contrasting several others, which is not new). Novel information is related to the “mild” pHPT category and subcategories, which is interesting (“normohormonal” subset). The sample size is more than adequate, however, the methods (including patient selection for imaging and statistics) require clarification or modification, as well as the interpretation of the results and the conclusion in particular in the light of the already existing literature. 

Specific comments

Introduction:

  • “...Approximately half of mPHPT patients and two-thirds of NHmPHPT cases experience at least one symptom[5, 9]…”: how does this correlate to the definition of mPHPT, stated as “...resulted in the identification of asymptomatic patients with non-specific biochemical results[5]…”? Explain!

  • “...we aimed to evaluate the diagnostic performance of [18F]FCH PET/CT as the first-line preoperative imaging in a large patient population with clinically suspicious PHPT…”: why perform diagnostic imaging in “clinically suspicious” circumstances? The indication is usually a planned surgical procedure in the setting of biochemically confirmed pHPT; explain!

Materials, Methods

  • “Exclusion criteria were the initial diagnosis of secondary/tertiary hyperparathyroidism (n=11), vitamin D deficiency (n=4), and familial hypocalciuric hypercalcemia (n=4).”: apart from s/tHPT, what was the indication for imaging? One would expect exclusion of these prior to imaging? Explain or correct!

  • “The semi-quantitative parameters, including maximum standard uptake value (SUVmax), average SUV (SUVmean), peak and average SUV normalized by lean body mass (SULpeak and SULmean, respectively), were measured within a selected iso-contour sphere as the volume of interest (VOI). The maximum metabolic short-axis diameters (mSAD) of the abnormal parathyroid lesion were determined on the PET images.”: was a specific threshold value used for isocontouring? Which? This will also significantly determine the measurement/values of the majority of the metrics (apart form SUV max) as described – how was this influence addressed? Explain!

  • “A successful resection was considered as >50% intraoperative iPTH decline compared to the baseline iPTH and decreasing pattern of iPTH and calcium in the laboratory follow-ups within six months after surgery.”: please provide more details on the surgical procedure (I assume minimally invasive approach was used, but please clarify).

  • “In the group of oPHPT patients who underwent the conservative approach, imaging and/or clinical follow-ups were considered the reference standard.” and “Clinical and imaging follow-ups were considered the reference standard in the group of mPHPT patients who underwent the conservative approach.”: again, why was imaging performed in these patiens, and repeated imaging in particular, if surgery was not considered an initial or delayed option? Explain!

  • “According to the reference standard, the receiver operating characteristic (ROC) curves fitted for sensitivity and specificity were drawn. Also, the area under the curve (AUC) was calculated.”: it makes sense to construct the ROC curves to characterize a specific test with a range of cut-off values for optimal Sn and Sp; this is not the case here (a simple positive-negative dichotomy) so ROC curve and the AUC does not provide significant additional information. These can be removed.

Results

  • “Two other patients had positive [18F]FCH PET/CT results, but the available pathology reports were negative for HFPT. These patients showed a more than 50% decrease in iPTH during the surgery, and iPTH remained in the average range years after the surgery.”: Average? Does this mean normal or reference range? Explain or correct.

  • remove ROC curves (detailed above).

Discussion

  • “Only a few published studies have evaluated [18F]FCH PET/CT as the first-line imaging modality in PHPT.”: this is not true – several studies have addressed this issue with [18F]FCH PET/CT performed as a first line modality, typically in parallel with [99mTc]Tc-sestaMIBI planar/SPECT/CT imaging or even subtraction imaging (at least Lezaic et al. 2014; Thanseer et al. 2017, Hocevar et al. 2017, Beheshti et al. 2018, Cuderman et al. 2020, in addition to Wouter et al. 2019 as an exclusive imaging modality) – all confirming the far superior diagnostic performance of [18F]FCH PET/CT over conventional imaging. Correct and discuss accordingly.

  • “The impact of [18F]FCH PET/CT in detecting MGD and ectopic parathyroid adenomas has been discussed controversially in the literature. Most studies reported promising results emphasizing the potential of [18F]FCH PET/CT to overcome the limitations of other modalities, as well as the determination of the lipomatous parathyroid adenomas [17, 28, 43, 44]. In contrast, some studies revealed a higher probability of FP findings on [18F]FCH PET/CT imaging [33, 45].”: Is this true? at least several papers from the same group (Lezaic et al 2014, Cuderman et al. 2020; several others as well) have systematically analyzed and shown the superior diagnostic performance of [18F]FCH PET/CT in MGD, in keeping with the current study – reevaluate and discuss.

Conclusion

  • [18F]FCH PET/CT can be potentially considered as first-line imaging for assessing PHPT. However, larger studies and cost-effectiveness investigations are mandatory to recommend [18F]FCH PET/CT as a routine modality in the clinic.”: What is the reasoning behind this statement? Although not completely standardized (perhaps even more for this reason!), [18F]FCH PET/CT has been performed in thousands of patients already with consistently impressive diagnostic performance and additional advantages over conventional imaging. It is considered as an “alternative” imaging approach that should be performed whenever possible in the current EANM guidelines! The only question remaining is the cost-effectiveness of the procedure; as already suggested before (Bossert et al. 2019), it is already considered as cost effective, and there are several works completed (Yap et al. 2021 – less adaptable to European context) or in progress (Quak et al. APACH2, 2021) that should provide the answer to this question. Add, correct, discuss!

Author Response

General comments

The authors retrospectively review a large patient cohort of patients with primary hyperparathyroidism (pHPT) that underwent [18F]fluorocholine ([18F]FCH) PET/CT as a first-line imaging for localization of hyperfunctioning parathyroid tissue. The patients were subdivided into the “overt” and “mild” pHPT categories according to symptoms and laboratory results; the “mild” pHPT group was further subdivided into the “normocalcemic” and “normohormonal” categories. Diagnostic peroformance of [18F]FCH PET/CT was evaluated against post-op histology as a standard of truth; interestingly, a number of patients was managed conservatively without referral to surgery with F/U w/imaging and lab data as a standard of truth. Imaging-derived metabolic parameters were correlated to laboratory results. [18F]FCH PET/CT was shown to have an impressive diagnostic performance in both “overt” and “mild” pHPT groups with the derived metabolic parameters significantly correlated to some biochemistry data.

Overall, the study confirms the already known excellent diagnostic performance of [18F]FCH PET/CT in pHPT; the relation between biochemistry values and (most of) the metabolic metrics is in keeping with several previous works (and contrasting several others, which is not new). Novel information is related to the “mild” pHPT category and subcategories, which is interesting (“normohormonal” subset). The sample size is more than adequate, however, the methods (including patient selection for imaging and statistics) require clarification or modification, as well as the interpretation of the results and the conclusion in particular in the light of the already existing literature. 

A: Thanks, dear reviewer. We firmly believe your comments enhanced our manuscript significantly.

Specific comments

Introduction:

  • “...Approximately half of mPHPT patients and two-thirds of NHmPHPT cases experience at least one symptom[5, 9]…”: how does this correlate to the definition of mPHPT, stated as “...resulted in the identification of asymptomatic patients with non-specific biochemical results[5]…”? Explain!

A: Thanks for your comment. We rewrote the sentences to deliver our message clearly.

  • “...we aimed to evaluate the diagnostic performance of [18F]FCH PET/CT as the first-line preoperative imaging in a large patient population with clinically suspicious PHPT…”: why perform diagnostic imaging in “clinically suspicious” circumstances? The indication is usually a planned surgical procedure in the setting of biochemically confirmed pHPT; explain!

A: Thanks for your comment. By saying “clinically suspicious” we wanted to distinguish it from the “confirmed” disease. We changed it to “clinically diagnosed” to address this problem.

Materials, Methods

  • “Exclusion criteria were the initial diagnosis of secondary/tertiary hyperparathyroidism (n=11), vitamin D deficiency (n=4), and familial hypocalciuric hypercalcemia (n=4).”: apart from s/tHPT, what was the indication for imaging? One would expect exclusion of these prior to imaging? Explain or correct!

A: Well, honestly, we agree with you. However, this limitation is mainly due to the retrospective design of this study. These 8 patients were found to be Vit D deficient or having FHH after further investigation. We explained it shortly.

  • “The semi-quantitative parameters, including maximum standard uptake value (SUVmax), average SUV (SUVmean), peak and average SUV normalized by lean body mass (SULpeak and SULmean, respectively), were measured within a selected iso-contour sphere as the volume of interest (VOI). The maximum metabolic short-axis diameters (mSAD) of the abnormal parathyroid lesion were determined on the PET images.”: was a specific threshold value used for isocontouring? Which? This will also significantly determine the measurement/values of the majority of the metrics (apart form SUV max) as described – how was this influence addressed? Explain!

A: Thanks for your comment. We added the predefined threshold in the software. Since that is routine thresholding in the clinic, we think of it as not a problem to be addressed. Hope you agree with us.

  • “A successful resection was considered as >50% intraoperative iPTH decline compared to the baseline iPTH and decreasing pattern of iPTH and calcium in the laboratory follow-ups within six months after surgery.”: please provide more details on the surgical procedure (I assume minimally invasive approach was used, but please clarify).

A: Exactly as you assumed! We clarified it.

  • “In the group of oPHPT patients who underwent the conservative approach, imaging and/or clinical follow-ups were considered the reference standard.” and “Clinical and imaging follow-ups were considered the reference standard in the group of mPHPT patients who underwent the conservative approach.”: again, why was imaging performed in these patiens, and repeated imaging in particular, if surgery was not considered an initial or delayed option? Explain!

A: Well, considering the conservative approach was related to multiple factors, such as the general status of the patients, the patient’s decision, imaging findings, etc. at the time of imaging. These patients either suffered from a symptom or presented again with abnormal lab tests in the follow-ups. Therefore, the referring physicians might want to reconsider the surgical option as a therapeutic option, and get reassurance about the “final diagnosis”, or guess a false-negative result in the previous imaging session, or assume that during follow-up, the lesion might grow and can now be visualized.

  • “According to the reference standard, the receiver operating characteristic (ROC) curves fitted for sensitivity and specificity were drawn. Also, the area under the curve (AUC) was calculated.”: it makes sense to construct the ROC curves to characterize a specific test with a range of cut-off values for optimal Sn and Sp; this is not the case here (a simple positive-negative dichotomy) so ROC curve and the AUC does not provide significant additional information. These can be removed.

A: Thanks for your comment. Based on your suggestion, we removed the ROC analysis. However, we kept the ROC analysis for the comparison between PET/CT and US since we think it can visualize the difference for the readers. Hope you agree with us.

Results

  • “Two other patients had positive [18F]FCH PET/CT results, but the available pathology reports were negative for HFPT. These patients showed a more than 50% decrease in iPTH during the surgery, and iPTH remained in the average range years after the surgery.”: Average? Does this mean normal or reference range? Explain or correct.

A: You are right! We corrected the vague info.

  • remove ROC curves (detailed above).

A: Again, we only kept the second ROC figure to depict the difference. Other ROC results were removed. Thanks!

Discussion

  • “Only a few published studies have evaluated [18F]FCH PET/CT as the first-line imaging modality in PHPT.”: this is not true – several studies have addressed this issue with [18F]FCH PET/CT performed as a first line modality, typically in parallel with [99mTc]Tc-sestaMIBI planar/SPECT/CT imaging or even subtraction imaging (at least Lezaic et al. 2014; Thanseer et al. 2017, Hocevar et al. 2017, Beheshti et al. 2018, Cuderman et al. 2020, in addition to Wouter et al. 2019 as an exclusive imaging modality) – all confirming the far superior diagnostic performance of [18F]FCH PET/CT over conventional imaging. Correct and discuss accordingly.

A: We agree with you. So, we edited the incorrect statement.

  • “The impact of [18F]FCH PET/CT in detecting MGD and ectopic parathyroid adenomas has been discussed controversially in the literature. Most studies reported promising results emphasizing the potential of [18F]FCH PET/CT to overcome the limitations of other modalities, as well as the determination of the lipomatous parathyroid adenomas [17, 28, 43, 44]. In contrast, some studies revealed a higher probability of FP findings on [18F]FCH PET/CT imaging [33, 45].”: Is this true? at least several papers from the same group (Lezaic et al 2014, Cuderman et al. 2020; several others as well) have systematically analyzed and shown the superior diagnostic performance of [18F]FCH PET/CT in MGD, in keeping with the current study – reevaluate and discuss.

A: It seems that we generalized a limited truth. The changes were made.

Conclusion

  • “[18F]FCH PET/CT can be potentially considered as first-line imaging for assessing PHPT. However, larger studies and cost-effectiveness investigations are mandatory to recommend [18F]FCH PET/CT as a routine modality in the clinic.”: What is the reasoning behind this statement? Although not completely standardized (perhaps even more for this reason!), [18F]FCH PET/CT has been performed in thousands of patients already with consistently impressive diagnostic performance and additional advantages over conventional imaging. It is considered as an “alternative” imaging approach that should be performed whenever possible in the current EANM guidelines! The only question remaining is the cost-effectiveness of the procedure; as already suggested before (Bossert et al. 2019), it is already considered as cost effective, and there are several works completed (Yap et al. 2021 – less adaptable to European context) or in progress (Quak et al. APACH2, 2021) that should provide the answer to this question. Add, correct, discuss!

A: Wow! Thanks for the points. Now we are on the same page as we found the mentioned point one hundred percent correct. We rewrote the conclusion to be more accurate regarding the application of FCH PET/CT in the clinic. Thanks again.

Reviewer 2 Report

In this retrospective study, the Authors evaluate the diagnostic performance of [18F]FCH PET/TC in a cohort of patients with PHPT who underwent [18F]FCH PET/TC as first-line imaging.

The study is interesting but the article needs of some clarifications.

11.       In the abstract (page 1, line 5) the Authors write about patients with “clinically suspicious PHPT”; all the study makes sense if the patients have a biochemical “diagnosis” of PHPT and not only a “clinical suspicious”; please clarify.

22.       In the introduction (page 1 line 10), I suggest to modify “asymptomatic patients with non-specific biochemical results” with “asymptomatic patients with borderline biochemical results”

33.       In the references: Are the reference 5 and the reference 6 the same reference? If yes please select the correct one and consequently modify the list.

44.       In the introduction (page 2, line 34) the reference number should be 12 instead of 24.

55.       In Materials and Methods, reference standard (page 3), please specify better what do you mean for “imaging” and “clinical follow up” that were considered the “reference standards”; this concept is explained in table S1, but I think it could be reported also in the text.

66.       In materials and methods, reference standard (page 3, line 18 of the paragraph), please clarify what do you mean for “normalized iPTH/calcium”: do you really mean the iPTH/calcium ratio or do you mean “normalized iPTH and calcium”?

77.       In table 2 (page 4) the “initial calcium” of the normo-hormonal true-positive group is higher than the one of the overt true-positive group; please confirm the date and provide a comment. Indeed, in presence of a “normal” PTH value, it can be expected also a less elevated serum calcium level; for example, a paper of Hollowoa (Normocalcemic and Normohormonal Primary Hyperparathyroidism: Laboratory Values and End-Organ Effects. Otolaryngology–Head and Neck Surgery. 2021) that compared NCpHPT and NHpHPT to classic pHPT, found lower calcium levels in the NHpHPT group than in the classic pHPT.

88.       In Results, oPHPT population, patient-based analysis (page 5, line 6): the Authors report three patients with evidence of persistent disease on follow up after surgery. Did all these patients have positive PET/TC and positive pathology results? One of them was undertreated; please provide a comment of the two others. Were they collocated as false positive?

99.       In Supplementary materials, study population: the normal range for serum calcium level was considered 2.2-2.8; is the upper level provided a typo? Instead it is too height and not consistent with the patient data.

110.   In all the paper the unit for measuring serum calcium is expressed in mmol/dL. Is it a mistake? Do you mean mmol/L?

111.   Please provide a comment regarding “mild” hyperparathyroidism, and in particular the normo-hormonal hyperparathyroidism. In the current guidelines the normo-hormonal hyperparathyroidism is not considered as a different entity in respect to classic hyperparathyroidism. However, this distinction may be useful for the purpose of this article; indeed, some studies have shown that in normo-hormonal hyperparathyroidism, the morphological and functional techniques, less frequently clearly identify hyperfunctioning parathyroid. Although in this study can be useful to evaluate the performance of PET/TC separately in overt and “mild” hyperparathyroidism, the Authors should add a comment in the discussion session.

Author Response

In this retrospective study, the Authors evaluate the diagnostic performance of [18F]FCH PET/TC in a cohort of patients with PHPT who underwent [18F]FCH PET/TC as first-line imaging.

The study is interesting but the article needs of some clarifications.

A: Thanks, dear reviewer. Following your suggestions, we think the manuscript is clarified for its future readers.

  1. In the abstract (page 1, line 5) the Authors write about patients with “clinically suspicious PHPT”; all the study makes sense if the patients have a biochemical “diagnosis” of PHPT and not only a “clinical suspicious”; please clarify.

A: Thanks for the comment. We corrected the issue.

  1. In the introduction (page 1 line 10), I suggest to modify “asymptomatic patients with non-specificbiochemical results” with “asymptomatic patients with borderline biochemical results”

A: Thanks. Done.

  1. In the references: Are the reference 5 and the reference 6 the same reference? If yes please select the correct one and consequently modify the list.

A: Thanks for your attention. The problem is fixed.

  1. In the introduction (page 2, line 34) the reference number should be 12 instead of 24.

A: Thanks again.

  1. In Materials and Methods, reference standard (page 3), please specify better what do you mean for “imaging” and “clinical follow up” that were considered the “reference standards”; this concept is explained in table S1, but I think it could be reported also in the text.

A: Thanks for your comment. We added the requested points.

  1. In materials and methods, reference standard (page 3, line 18 of the paragraph), please clarify what do you mean for “normalized iPTH/calcium”: do you really mean the iPTH/calcium ratio or do you mean “normalized iPTH and calcium”?

A: Thanks for the point. The second one was the real meaning. We clarified it.

  1. In table 2 (page 4) the “initial calcium” of the normo-hormonal true-positive group is higher than the one of the overt true-positive group; please confirm the date and provide a comment. Indeed, in presence of a “normal” PTH value, it can be expected also a less elevated serum calcium level; for example, a paper of Hollowoa (Normocalcemic and Normohormonal Primary Hyperparathyroidism: Laboratory Values and End-Organ Effects. Otolaryngology–Head and Neck Surgery. 2021) that compared NCpHPT and NHpHPT to classic pHPT, found lower calcium levels in the NHpHPT group than in the classic pHPT.

A: We rechecked the data and the stated results were correct. It seems that because of their high serum Calcium levels, the physicians referred them to undergo PET/CT. Do you agree with us regarding this referral bias?

  1. In Results, oPHPT population, patient-based analysis (page 5, line 6): the Authors report three patients with evidence of persistent disease on follow up after surgery. Did all these patients have positive PET/TC and positive pathology results? One of them was undertreated; please provide a comment of the two others. Were they collocated as false positive?

A: Yes, dear reviewer. Regarding the two mentioned patients, it seems that maybe the histopathology findings were not accurately reported since the patients benefited from the surgery.

  1. In Supplementary materials, study population: the normal range for serum calcium level was considered 2.2-2.8; is the upper level provided a typo? Instead it is too height and not consistent with the patient data.

A: Thanks for your attention. That was a typo!

  1. In all the paper the unit for measuring serum calcium is expressed in mmol/dL. Is it a mistake? Do you mean mmol/L?

A: Thanks again. Corrected.

  1. Please provide a comment regarding “mild” hyperparathyroidism, and in particular the normo-hormonal hyperparathyroidism. In the current guidelines the normo-hormonal hyperparathyroidism is not considered as a different entity in respect to classic hyperparathyroidism. However, this distinction may be useful for the purpose of this article; indeed, some studies have shown that in normo-hormonal hyperparathyroidism, the morphological and functional techniques, less frequently clearly identify hyperfunctioning parathyroid. Although in this study can be useful to evaluate the performance of PET/TC separately in overt and “mild” hyperparathyroidism, the Authors should add a comment in the discussion session.

A: Thanks for the comment. A note was added to the discussion.
